# Thermal contrast enhancement predicts paradoxical heat sensation
Alexandra G. Mitchell [1] ✉, Jesper Fischer Ehmsen[1], Małgorzata Basińska [2], Arthur S. Courtin [1], Rebecca A. Böhme [1,3], Camila Sardeto Deolindo [1], Micah G. Allen[1,4], Kristian Sandberg[1,5] & Francesca Fardo[1,6] ✉

Paradoxical Heat Sensation (PHS) is the remarkable feeling of warmth or heat pain while the skin is cooling. Despite its initial documentation over 100 years ago, a unified explanation for this perplexing experience remains elusive. Here we apply contrast enhancement principles, known for their instrumental role in understanding visual illusions, to the domain of thermosensation. Contrast enhancement describes the amplification of two contrasting visual features, such as the enhanced perception of an edge between a light and dark bar. In thermosensation, this encompasses an enhancement of the difference between sequential warming and cooling of the skin, and is defined as the normalised difference between successive temporal warm and cold temperatures. Remarkably, thermal contrast predicts the occurrence of PHS. Our findings reveal compelling evidence supporting the role of thermal contrast in the generation of PHS, shedding light on its underlying mechanism and offering a framework for broader encoding principles in thermosensation and pain.

Among the various thermal sensations experienced by humans, Paradoxical Heat Sensation (PHS) stands out as a particularly intriguing phenomenon[1–4]. PHS refers to the illusory perception of warmth during the cooling of the skin and is typically perceived when warming and cooling are temporally alternated. It is during these dynamic thermal alternations that the human thermosensory system misinterprets a temperature change from warm to cool as warmth, heat or heat pain. Despite previous research efforts, the underlying mechanisms of this counterintuitive sensory experience have remained elusive. Clarifying these mechanisms is essential not only for advancing our understanding of PHS but also for uncovering the broader principles that govern human thermosensory perception.

Here, we demonstrate that PHS reflects a temporal contrast enhancement mechanism in the human thermosensory system. We propose that the dynamic interplay between subsequent warming and cooling on the skin elicits neural responses that amplify the perceived difference between these temperature sensations, culminating in the paradoxical experience of heat. Contrast enhancement refers to the phenomenon where boundaries between two contrasting features (for example, light and dark bars) are perceptually enhanced[5]. This mechanism is fundamental for improving the sensitivity and discriminability of sensory information across

various perceptual domains, including vision and audition[6,7]. In the visual domain, for instance, differences in luminance between spatially adjacent stimuli are accentuated by the nervous system to more effectively detect and process features of the sensory environment, such as boundaries or edges[5,6]. Visual gratings, which consist of spatial patterns of alternating light and dark bars, provide a clear example of this principle (Fig. 1A).

Analogously, the thermal sensory limen (TSL) task, which is typically employed to assess the presence of PHS, can be considered as a temporal counterpart to visual gratings within the thermosensory system. In the TSL, warming and cooling are alternated at the same skin location, with these temperature changes representing the approximate peak and trough of a sinusoidal function over time (Fig. 1B). A single trial of the TSL is defined as one thermal cycle where the temperature probe increases to the set maximum temperature, then decreases past baseline (32 °C) to the point at which the participant detects a change in sensation or pain, before returning to the baseline temperature. From this perspective, we defined the Thermal Contrast Function (see Eq. 1) as the normalised difference between subsequent warm and cold temperatures for each trial. This function yields a standardised value that captures the maximum and minimum temperatures of a single TSL trial and can be compared across different conditions, experiments, or patient populations.

[1]Center of Functionally Integrative Neuroscience, Department of Clinical Medicine, Aarhus University, Aarhus, Denmark. [2]Department of Psychology, Medical University of Gdansk, Gdansk, Poland. [3]Department of Psychology, Lund University, Lund, Sweden. [4]Cambridge Psychiatry, University of Cambridge, Cambridge, UK. [5]Aarhus University Hospital, Aarhus University, Aarhus, Denmark. [6]Danish Pain Research Center, Department of Clinical Medicine, Aarhus University, Aarhus, Denmark. ✉e-mail: agmitchell@cfin.au.dk; francesca@cfin.au.dk

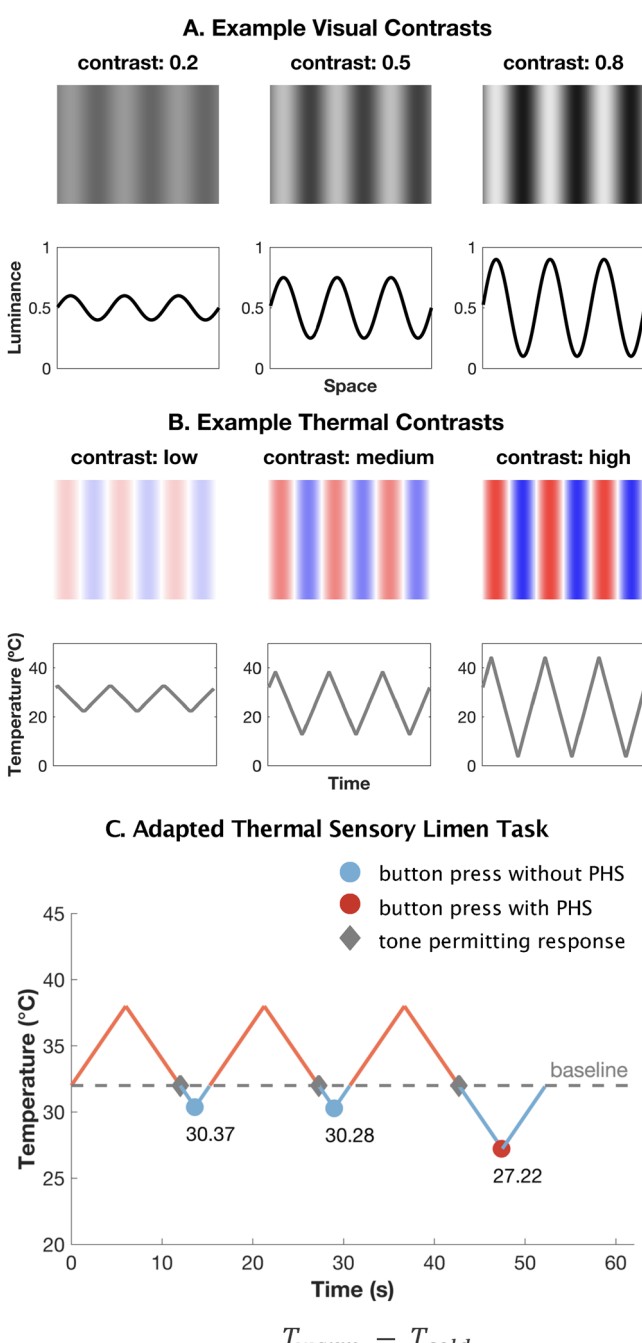

## A. Example Visual Contrasts

contrast: 0.2    contrast: 0.5    contrast: 0.8

## B. Example Thermal Contrasts

contrast: low    contrast: medium    contrast: high

## C. Adapted Thermal Sensory Limen Task

● button press without PHS
● button press with PHS
◆ tone permitting response

$$TCF = \frac{T_{warm} - T_{cold}}{T_{\max warm} - T_{\min cold}}$$

**Fig. 1 | Contrast in the thermosensory domain. A** Visual gratings and corresponding sinusoidal waveforms representing three different contrasts; low (0.20), medium (0.50) and high (0.80). The corresponding sinusoidal waveforms have a relatively low, intermediate and high amplitude, reflecting an increasing change in luminance and increasingly sharpened difference between light and dark bars. **B** Example temporal profiles (warm = red, cold = blue) and their corresponding Thermal Sensory Limen profiles for low, medium and high thermal contrast that match the corresponding visual contrast level shown in (**A**). **C** An example of the adapted thermal sensory limen task sequence from one participant shows three experimental trials for the medium contrast condition, with peaks at a starting temperature of 38 °C and troughs at the point at which they detected a change in sensation. An auditory tone was presented when the temperature returned from peak to 32 °C, which corresponded to the beginning of the cooling phase. Participants pressed a button at perceived temperature changes (innocuous condition) or pain (noxious condition), followed by a description of the sensation (cold, warm, cold pain or heat pain). The TSL threshold corresponded to the temperature of the probe when the button was pressed. TCF thermal contrast function is used to calculate contrast for each TSL trial where 50 °C is the maximum possible temperature (*Tmax warm*), and 0 °C is the minimum possible temperature (*Tmin warm*).

The primary outcome measures were the probability of innocuous and noxious PHS occurring on a specific trial, and the associated thermal thresholds. For Hypothesis 1, the rate of PHS for each contrast condition was also used. Thresholds were the temperature at which participants noticed a sensation change (innocuous TSL threshold) or pain (noxious TSL threshold). Innocuous PHS was the reported experience of warmth, while noxious PHS was the reported experience of heat pain during the cooling phase of the TSL. These two tasks were chosen to explore the differences between PHS elicited from innocuous and noxious thermal stimuli. To specifically address the role of contrast enhancement, we calculated the thermal contrast (Fig. 1C, Eq. 1) for each trial in low, medium and high contrast conditions.

## Methods
### Participants
Participants were recruited through the Centre of Functionally Integrative Neuroscience, Aarhus University and tested as part of the EU COST Action CA18106 between August 2019 and March 2021. The data presented in this manuscript constitutes part of a larger data set of behavioural and MRI data from healthy participants, and the study was not pre-registered. Recruitment criteria were healthy participants between the ages of 18 and 50 with no reported history of neurological illness damage or pain disorders. We also ensured that participants did not have skin conditions (such as eczema), scars, tattoos on their dorsal forearms prior to data collection, or any other conditions that may affect the experience of temperature or pain at this site. Participants were compensated 125 DKK for their time.

A total of 215 participants took part in the study, with seven participants excluded from further analysis due to missing trial data. The final sample consisted of 208 participants, consisting of 121 women (gender determined through self-report), mean age = 24.97 years old (SD = 5.19, range: 18–49). Race or ethnicity was not recorded. This study was conducted in accordance with the Declaration of Helsinki and was approved by the Local Ethical Committee of Region Midtjylland, Denmark. Informed consent was obtained from all participants prior to the start of the study, and all relevant ethical regulations were followed. A post-hoc power determination analysis showed that our final sample size provided over 90% power for logistic regression to detect a two-tailed effect with an odds ratio of at least 1.5 for a predictor with lognormal distribution and at least 80% power for a predictor with a normal distribution.

### Tasks
The experimental paradigm involved placing a single (32 × 32 mm) peltier-based contact thermode (TSA stimulator, Medoc Advanced Medical

To investigate the role of contrast enhancement in PHS, we conducted a study of 208 healthy participants using an adapted TSL task featuring low, medium and high contrast conditions. These conditions were established by variations between a fixed maximum temperature, corresponding to the peak in the warm temperature range (at 32, 38, and 44 °C) and a variable minimum temperature, corresponding to the trough in the cold temperature range. This variable minimum temperature was determined by participants' responses based on two task instructions. Participants pressed a button upon experiencing either a change in sensation (innocuous task condition) or a painful sensation (noxious task condition). Each button press led to a reversal in the temperature fluctuation. Responses were permitted only when the probe's temperature entered the cold temperature range, below 32 °C to a minimum of 0 °C, set for safety purposes. An auditory tone signalled the start of the response window (Fig. 1C).

Systems) on the dorsal surface of the participant's forearm and assessing thermal thresholds across both innocuous and noxious temperature ranges. We measured cold and warm thermal thresholds (i.e., detection, TSL and pain thresholds) following the procedure and instructions provided in the German Research Network on Neuropathic Pain (DFNS) quantitative sensory testing (QST) battery[8–10].

Subsequently, participants completed an adapted version of the TSL task[11], where we defined three levels of thermal contrast by controlling the warm starting temperature prior to each trial. In this task, participants were instructed to press a button as soon as they experienced a specific change in sensation during the cooling phase of a single thermal probe. The temperature of the probe ramped down, at a rate of 1 °C/s, from one of three possible starting temperatures that defined our contrast conditions (32, 38 or 44 °C), eliciting a neutral, warm, or hot sensation, respectively. This differed from the standard TSL task, where the peak temperatures are not fixed and are dependent on the participant experiencing a change in sensation during the heating phase in each trial. Once the probe reached the baseline temperature of 32 °C, an auditory tone was played to mark the start of the trial, prompting the participant to provide a response based on the given instructions. This is another addition to the classic TSL paradigm to ensure that the probe was in the cooling phase (i.e. below baseline temperature) before the participants judged a temperature change. Participants were informed to press a button when they experienced any change in sensation (innocuous condition) or a painful sensation (noxious condition) and verbally report the quality of the sensation they experienced (e.g., cold, warm, cold pain or heat pain). The innocuous and noxious task conditions were developed to distinguish between PHS elicited from innocuous and noxious temperatures. They were also defined to represent the quality of stimuli the participant should be experiencing in the absence of any PHS and to clearly distinguish TSL thresholds with cold and warm thermal detection and cold and heat pain thresholds defined through QST. If the starting temperature was set to baseline, 32 °C, the tone was played after two seconds of baseline temperature stimulation. After participants had pressed the button, the temperature of the probe increased at a rate of 1 °C/s until it reached the fixed starting temperature (32, 38 or 44 °C), at which point the temperature decreased again, creating a temporal pattern of alternated warming and cooling of the skin (Fig. 1C).

## Procedure

Participants completed six trials (three practice, three experimental) for each contrast (32, 38 and 44 °C) and task condition (innocuous and noxious), resulting in a total of 36 TSL trials per participant. This trial structure reflects a standard TSL protocol. A single trial consists of the temperature probe starting at baseline, then increasing to the set maximum temperature before decreasing past baseline (32 °C). At this point a tone was played to indicate the probe was in the cooling phase of the TSL. The temperature of the probe decreased until the point at which the participant detected a change in sensation or pain and then returned to the baseline temperature. The probe was in contact with the skin for all six trials in each condition. To prevent carry-over effects from intense thermal stimulation, the location of the thermal probe on the dorsal forearm was moved every six trials, which corresponded to a different stimulation site on the forearm for every starting temperature. The order of task conditions and starting temperatures was the same for all participants. Participants completed the innocuous TSL condition first before the noxious TSL condition. In both innocuous and noxious conditions, the first starting temperature was 32 °C, followed by 38 °C and then 44 °C. Task and trial orders were specifically chosen to reduce the likelihood of carryover effects from noxious stimulation of the skin into subsequent trials and dependent variables were extracted from the three experimental trials only.

## Variables

Dependent variables of interest were the number of trials where a paradoxical heat sensation (PHS) occurred, the probability of PHS occurring in a specific trial, and thermal sensory limen (TSL) thresholds for each trial. A PHS was defined as the perception of warmth or heat (innocuous condition) or heat pain (noxious condition) during the cooling of the skin. TSL thresholds corresponded to the temperature on the skin at the time of the participant's response and reflected the temperature at which participants experienced a change in thermal sensation (innocuous condition) or a painful percept (noxious condition) during the cooling of the skin. Thermal contrast was quantified for each trial in each condition, in each participant using the Thermal Contrast Function (TCF, Eq. 1, Fig. 1C), which divides the difference between the warm and cold temperatures in each trial by the difference between the max and min temperature cut-offs (i.e., 50–0 °C).

$$\text{TCF} = \frac{T_{\text{warm}} - T_{\text{cold}}}{T_{\text{max warm}} - T_{\text{min cold}}} \qquad (1)$$

## Statistics and reproducibility

All tests performed were two-tailed with an alpha criterion of 0.05 and conducted on the entire participant sample ($n = 208$).

The association between thermal contrast and the prevalence of PHS was assessed using three McNemar's tests (32° vs 38°, 38° vs 44° and 32° vs 44°), to determine whether the distribution of individuals with PHS increased with increasing contrast level. Three McNemar's tests were conducted for each contrast condition in each task (innocuous and noxious). Next, we conducted one mixed-effect logistic regression model with fixed effects of thermal contrast (low 32 °C, medium 38 °C and high 44 °C) and task (innocuous and noxious) and random effect of participant ID to test whether thermal contrast predicted the prevalence of innocuous and noxious PHS (Model 1A). To address the possibility that PHS presence is modulated by trial number, an additional model including trial number as a fixed effect (Model S1) can be seen in Supplementary Note 1. Due to the absence of a clear and convincing relationship between contrast conditions and noxious PHS, we conducted the following analyses on PHS that occurred during the innocuous TSL only.

$$\text{PHS} \sim \text{contrastCondition} * \text{task} + (1|\text{ID}) \qquad (\text{Model1A})$$

To test the effect of contrast (starting temperature) on TSL threshold temperatures during innocuous and noxious conditions, we used two mixed-effect linear regression models (Models 2A and 2B), each with a fixed effect of contrast condition and random intercept of participant ID. Next, we ran a mixed-effect logistic regression model with fixed effects of normalised (z-scored) innocuous and noxious TSL thresholds to assess the relationship between TSL thresholds and PHS prevalence (Model 2C).

$$\text{innocuousTSL} \sim \text{contrastCondition} + \text{trial} + (1|\text{ID}) \qquad (\text{Model2A})$$

$$\text{noxiousTSL} \sim \text{contrastCondition} + \text{trial} + (1|\text{ID}) \qquad (\text{Model2B})$$

$$\text{innocuousPHS} \sim \text{innocuousTSL} * \text{noxiousTSL} + (1|\text{ID}) \qquad (\text{Model2C})$$

To test whether thermal contrast predicts PHS prevalence, we modelled trial-by-trial thermal contrast (TCF) values in three mixed-effect logistic regression models (Models 3A–C). Fixed effects included the innocuous thermal contrast for each trial, the mean noxious thermal contrast across trials, and a random intercept of participant ID. Both innocuous and noxious TCF values were transformed using log10 normalisation. To compare performance between the contrast models to predict PHS, the areas under (AUC) receiver operator characteristic (ROC) curves were used to calculate model accuracy. To assess this uncertainty around AUC estimates, data were resampled, using the *rsample* package in R[12], into training and test data sets, both of the same size as the original data. Each model was repeatedly ($n = 2000$) fitted on a training data set, and the performance of the model was evaluated on a test data set.

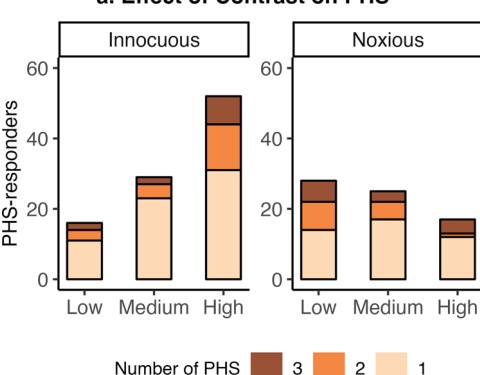

### a. Effect of Contrast on PHS

Number of PHS: 3, 2, 1

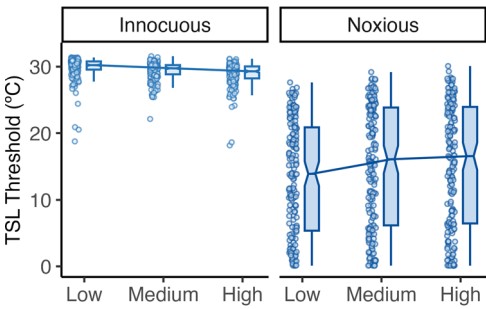

### b. Individual TSL Thresholds

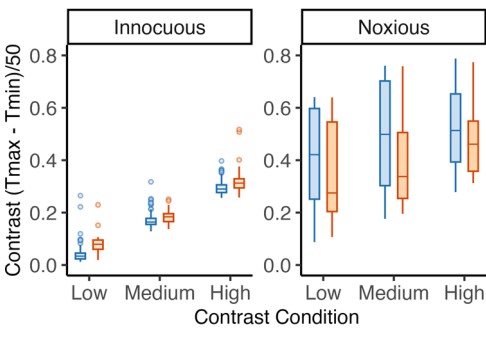

### c. Thermal Contrast by PHS

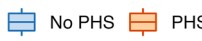

No PHS · PHS

**Fig. 2 | The effect of thermal contrast on TSL thresholds and PHS. A** The overall rate of innocuous PHS increases with increasing contrast, whilst contrast does not affect noxious PHS. PHS prevalence (number of participants with PHS) and rate (the total number of PHS over three trials) are shown for each contrast condition and task. **B** Thermal contrast also affects TSL thresholds in both task conditions. TSL temperatures at which participants detected a change in sensation (innocuous) decrease with increasing contrast, whilst TSL temperatures at which participants detect a painful sensation (noxious) increase with increasing contrast. Box plots show the median with confidence intervals (notches), and the dots show the individual participant's mean. Noxious TSL thresholds below .10 °C were removed for visualisation purposes only. **C** Thermal contrast defined by Eq. 1 (the Thermal Contrast Function) increases with contrast condition (starting temperature) but is higher in individuals who experience PHS for the innocuous condition and lower in the noxious condition, compared to those participants who experience veridical cold.

Bootstrapped samples were then used to model 95% confidence intervals fitted AUCs for each logistic regression model. This approach was chosen to test model accuracy over cross-validated n-fold to reduce the likelihood of over-inflation of accuracy due to the rare prevalence of PHS in certain conditions (e.g., the number of innocuous PHS in the low contrast

condition was 17/208).

$$\text{innocuousPHS} \sim \text{innocuousTCF} * \text{noxiousTCF} + (1|\text{ID}) \quad \text{(Model3A)}$$

$$\text{innocuousPHS} \sim \text{innocuousTCF} + (1|\text{ID}) \quad \text{(Model3B)}$$

$$\text{innocuousPHS} \sim \text{noxiousTCF} + (1|\text{ID}) \quad \text{(Model3C)}$$

To determine whether PHS is related specifically to thermal contrast during TSL over detection thresholds defined in the QST protocol, a final mixed-effect logistic regression model was conducted with warm and cold detection and pain thresholds measured through QST as fixed effects and participant ID as a random intercept. In addition to this, we added age and gender as predictors to models 2A, 2B and 3A to test whether age and gender of the participants affect the probability of PHS. The results of these models are presented in Supplementary Tables 1–12. Finally, models (S2–S4) that include the effect of age and gender on both TSL thresholds and PHS are reported in Supplementary Note 2.

### Reporting summary

Further information on research design is available in the Nature Portfolio Reporting Summary linked to this article.

## Results

### Contrast is associated with both PHS and innocuous and noxious TSL thresholds

We first investigated whether the rate of paradoxical heat sensation depended upon contrast level, defined by the low, medium and high contrast conditions. To this end, we compared the distribution of PHS (total no-PHS and PHS count) occurrences across the three contrast levels (32, 38 and 44 °C) and two TSL tasks (innocuous vs. noxious) using six McNemar's Chi-Squared tests (Fig. 2A). As contrast increased, the number of individuals reporting innocuous PHS increased from low to medium contrasts (32–38 °C, $\chi^2(1) = 4.36$, $p = 0.04$, Cohen's $g = -0.20$), low to high contrasts (32–44 °C, $\chi^2(1) = 23.56$, $p < 0.001$, Cohen's $g = -0.35$) and medium to high contrasts (38–44 °C, $\chi^2(1) = 9.88$, $p = 0.002$, Cohen's $g = -0.24$). The number of individuals reporting noxious PHS did not exhibit a clear relationship with the increase in contrast (32° vs. 38° $\chi^2(1) = 0.11$, $p = 0.62$, Cohen's $g = 0.04$; 32° vs. 44° $\chi^2(1) = 2.70$, $p = 0.10$, Cohen's $g = 0.15$; 38° vs. 44 ° $\chi^2(1) = 1.75$, $p = 0.19$, Cohen's $g = 0.14$).

In addition to this, a mixed-effect regression model testing the probability of PHS in each trial showed significant interaction effects between task and contrast conditions (Model 1A). This was confirmed through a follow-up omnibus Chi-squared test ($\chi^2(2) = 40.47$, $p < 0.001$, Supplementary Table 2) and post-hoc pairwise comparisons showed that the probability of innocuous PHS occurring within a specific trial increased with increasing contrast between the medium and high (44–38 °C, $\beta = -1.05$, 95% CI −1.49 to −0.60, $p < 0.001$, $z$-ratio = −4.58) and low and high contrast conditions (44–32 °C, $\beta = -1.62$, 95% CI −2.13 to −1.09, $p < .001$, $z$-ratio = −6.11), but not between low and medium contrasts (38–32 °C, $\beta = -0.57$, 95% CI −1.15 to 0.00, $p = 0.12$, $z$-ratio = −1.97). The probability of noxious PHS did not significantly change between low and medium (32 vs. 38 °C, $\beta = 0.36$, 95% CI −0.12 to 0.85, $p = 0.31$, $z$-ratio = −1.47) or medium and high (38 vs. 44 °C, $\beta = 0.39$, 95% CI −0.16 to 0.95, $p = 0.34$, $z$-ratio = −1.39) contrast conditions, but significantly decreased between the low and high contrasts (32 vs. 44 °C, $\beta = 0.76$, 95% CI 0.23–1.29, $p = 0.01$, $z$-ratio = −2.81). These results indicate that the prevalence and rate of innocuous, but not noxious, PHS is increased by contrast created by the temporal alternation of warming and cooling of the skin.

We next analysed whether the threshold temperatures at which individuals experienced a change in sensation or a painful sensation varied as a function of contrast (Fig. 2B). Innocuous TSL thresholds increased from low to medium ($t(1661.00) = -4.86$, $\beta = -0.46$, 95% CI −0.65 to −0.28, $p < 0.001$) and low to high contrasts ($t(1661.00) = -9.95$, $\beta = -0.95$, 95% CI −1.14 to −0.76, $p < 0.001$, Model 2A), while noxious TSL thresholds

decreased with increasing contrast level (low to medium: $t(1660.99) = 9.05$, $\beta = 1.86$, 95% CI 1.45–2.26, $p < 0.001$; low to high: $t(1660.99) = 10.36$, $\beta = 2.12$, 95% CI 1.72–2.53, $p < 0.001$, Model 2B). We further found, using a logistic regression model, that the probability of innocuous PHS occurring in each trial increased significantly with increasing innocuous TSL thresholds ($z = -14.18$, $p < 0.001$, OR = 0.35, 95% CI = 0.26–0.45), and increased with decreasing noxious TSL thresholds (decreasing thresholds = higher temperatures produce cold pain, $z = 4.37$, $p = 0.001$, OR = 2.12, 95% CI = 1.44–3.14, Supplementary Fig. 1, Model 2C). Note that increasing thresholds for cold detection means participants require lower temperatures to detect cold, and decreasing thresholds mean that higher temperatures are required. This result indicates both innocuous and noxious thermal thresholds are related to the presence of PHS and are jointly driven by thermal contrast.

## PHS relates to temporal contrast enhancement in the thermosensory system

To demonstrate further evidence for our thermal contrast model, we quantified the relationship between temporal thermal contrast and individual TSL thresholds in each trial using an equation adapted from vision science. This equation defines the thermal contrast as the normalised difference between the maximum and minimum temperatures for each trial of the TSL (Thermal Contrast Function, Eq. 1). The distribution of TCF values for each condition can be observed in Supplementary Fig. 2. Alongside this, we ran logistic regression models to determine the predictive value of thermal contrast on PHS (Models 3A–C). See Methods for full model equations and Supplementary Tables 1–12 for full results from all four models.

The odds of innocuous PHS occurring in each trial increased with increasing innocuous thermal contrast (Fig. 2C, $z = 5.35$, $p < 0.001$, OR = 13.48, 95% CI 5.20–34.99). To understand the sensitivity vs specificity of this effect, we calculated the accuracy under (AUC) receiver operating characteristic (ROC) curves for four logistic regression models, including thermal contrast (Fig. 3). ROC curves indicated that a model that included innocuous and noxious TCF (Fig. 3B, AUC = 0.76, AIC = 791) predicted the probability of an innocuous PHS more accurately than three other comparison models that included contrast condition (Fig. 3A), innocuous TCF only (Fig. 3C), and noxious TSF only (Fig. 3C). The marginal effects of the simplest model (Fig. 3A) indicated a predictive probability of 6% in high contrast conditions, while in the winning model (Fig. 3D) this probability increased up to 75% for trials with a higher innocuous contrast and lower noxious contrast (measured using the TCF). While the occurrence of PHS is greatest for high contrast conditions, across all contrast levels, TCF values more accurately predict PHS probability.

As a control analysis, we aimed to confirm that PHS is driven by thermal contrast sensitivity specifically, rather than general detection and pain thresholds. We defined a logistic regression model to predict the likelihood of innocuous PHS from cold detection and cold pain thresholds obtained during the Quantitative Sensory Testing (QST) protocol. Our findings revealed that QST thermal thresholds were insufficient to predict PHS (Supplementary Tables 13 and 14). Therefore, computational modelling of PHS showed that thermal contrast created by the alternation of warming and cooling during the TSL specifically influences the probability of PHS in a given trial, above and beyond thermal sensitivity measures, such as cold detection and cold pain thresholds.

## Discussion

The study of illusions within the visual system has been essential in unravelling fundamental mechanisms of perception. Here, we capitalised on this to further our understanding of thermosensation through thermal nociceptive illusions. We assessed whether contrast enhancement, an established mechanism within the visual and auditory domains, is the driving mechanism behind PHS, using an adapted TSL task to create conditions of low, medium and high contrast. We found that increased thermal contrast during the TSL increases the number of PHS in our sample, highlighting the role of temporal contrast enhancement in the generation of paradoxical heat. The results of Model 3A showed that high thermal contrast during the TSL can increase the likelihood of a PHS trial occurring to a prevalence of more than 50%. By using a function to calculate thermal contrast for each TSL trial (the TCF), we were able to extract a single, standardised number to represent thermal contrast. This is a powerful approach, as it not only allows us to successfully define the relationship between PHS and temporal thermal contrast independently from other measures of thermal sensitivity but can also quantify patterns of thermal sensitivity among different populations and across datasets and methods that adopt different TSL procedures. As understanding illusions in other perceptual domains has fostered an increased understanding of complex sensory mechanisms, the computational approach motivated here should also inform our more general understanding of pain and thermosensory processes and their dysfunction in neurological diseases.

Prior research has overlooked the potential role of thermal contrast enhancement in PHS. This explanation could account for previously observed differences in PHS rates between healthy, young individuals[2,13,14], older individuals[8,11,13], as well as in patients with reduced thermosensory function due to peripheral and central nervous system disorders[3,13,15,16]. For example, PHS is relatively infrequent in healthy individuals but more prevalent in patients with neuropathy[17]. This discrepancy was identified in studies that used the traditional TSL task, where individual thermal sensitivity shapes the extent of skin warming and cooling (e.g., 8,13). As reduced thermal sensitivity is likely to lead to increased thermal contrast during TSL, it is, therefore, challenging to disentangle whether PHS is a pathological sign related to thermosensory loss or a result of known differences in TSL thresholds that naturally occur as a consequence of reduced thermal sensitivity in these groups. Our approach allows us to explicitly test the role of PHS, irrespective of thermosensory sensitivity and provides a standardised quantification of TSL outcomes that can be compared across groups.

Thermal contrast enhancement as a driving mechanism for PHS unifies previously incongruous findings. For example, after skin sensitisation using capsaicin, PHS responders exhibited higher warm and cold TSL thresholds and, therefore, increased thermal contrast during the TSL task, compared to non-responders[18]. Similarly, variations in warming and cooling patterns during TSL that affect both the number and intensity of PHS[2,11,14] can be reinterpreted using our current model. For example, previous findings showing increases in PHS after both pre-warming[14] and pre-cooling[11] can be explained as a consequence of an increase in temporal thermal contrast. In light of this, we argue that contrast enhancement provides a powerful explanation for the presence of PHS, and future research should carefully take into account the relation between thermal contrast and sensitivity in the manifestation of PHS.

## Limitations

The aim of our study was to clarify the relationship between the presence of PHS and thermal contrast, independent of individual thermal sensitivity during the TSL. Consequently, the fixed warm temperatures during the TSL task (32, 38 and 44 °C) did not account for individual differences in temperature and pain detection. As we are yet to develop a comprehensive understanding of the stimulus parameters required to produce PHS, the absence of PHS in some of our participants may, therefore, not be due to a lack of responsiveness to the illusion but instead because suboptimal parameters were used to assess PHS in those specific individuals. For example, 38 °C may be perceived as much less intense in an individual with low thermal sensitivity compared to an individual with high thermal sensitivity. Future studies should consider tailoring the temperatures to individual pain thresholds or using adaptive psychophysics to staircase specific parameters related to PHS experience. This will ensure that any manipulations to thermal contrast produce the desired increase or decrease in thermal intensity consistently across individuals. This approach is particularly important when considering group comparisons between healthy individuals and clinical populations with altered thermal sensitivity.

**Fig. 3 | ROC curves and predicted probability of PHS for each model showing that thermal contrast predicts PHS prevalence.** In the left panels ROC curves show the accuracy of each logistic regression model to predict innocuous PHS. The winning model is shown in panel (**B**). Coloured lines show the ROC for the data-driven model, whilst the grey lines show bootstrapped simulated ROCs ($n$ = 2000). The area under the curve (AUC) is displayed as an indicator of model accuracy, with 95% confidence intervals of bootstrapped AUCs indicated in square brackets. Model fit is indicated by AIC. The right panels show the predicted probability of the marginal effects of each model to lead to a PHS trial. In the $x$-axis, higher innocuous and noxious TCF (log10) represents higher contrast. The predicted probability of a PHS trial increases from around 5% in the simplest model (panel **A**) to up to 75% if innocuous and noxious TCF values are included in the model (panel **B**). Error bars (panel **A**) and shaded areas (panels **B**–**D**) show low and high confidence intervals for the marginal effects of each model predictor.

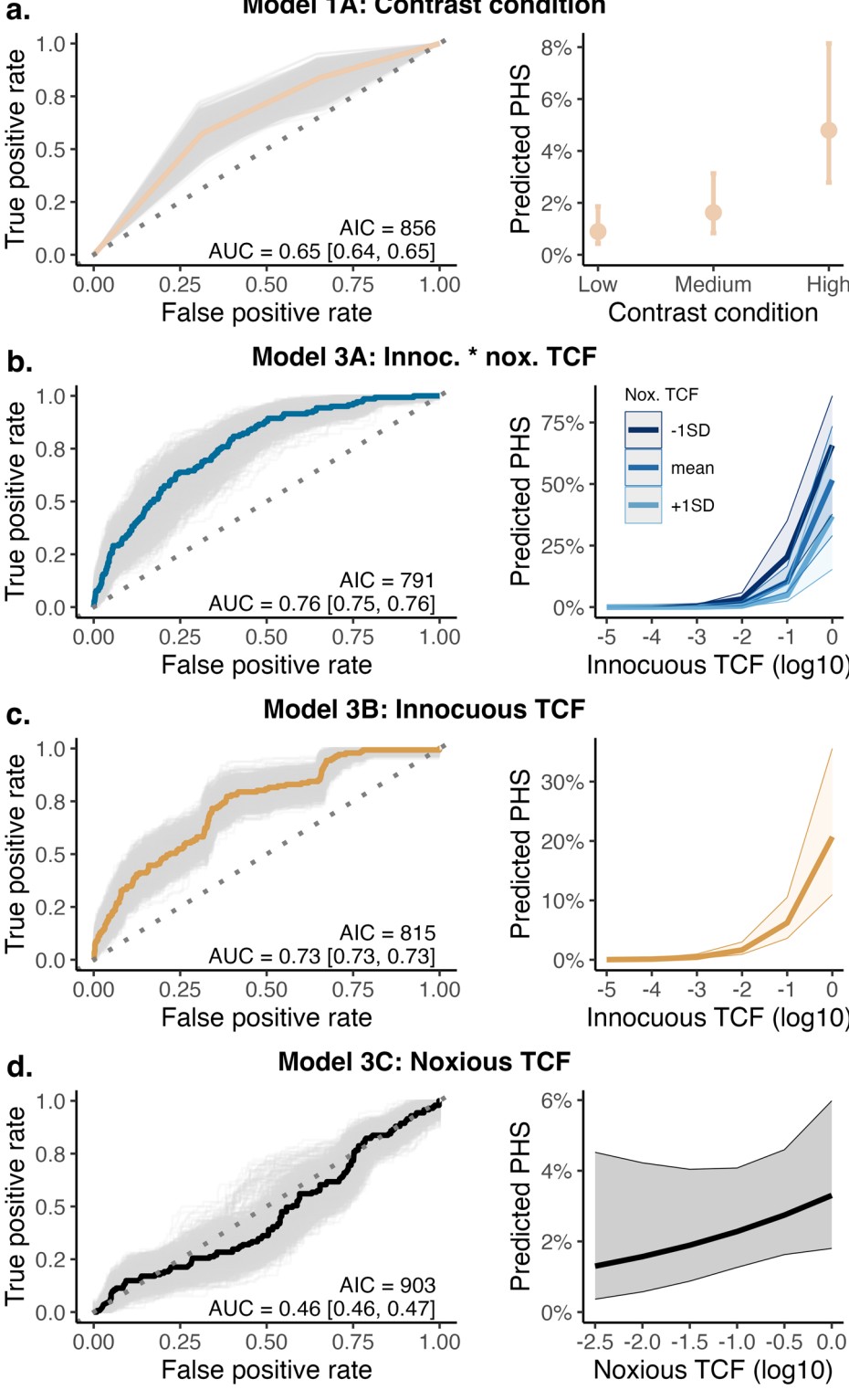

## Conclusion

Here, we capitalised on theories established through our understanding of illusions in the visual system to provide a unifying explanation for PHS, an illusion that has previously defied coherent explanations. Specifically, we proposed thermal contrast enhancement mechanisms underlie the experience of PHS. We used an adapted version of the TSL task to create conditions of low, medium and high contrast and developed a Thermal Contrast Function, which computes a single standardised number encoding the contrast on each TSL trial, for each individual. We established that increasing the thermal contrast, defined by a higher peak, during the TSL increased both the rate and prevalence of PHS. Alongside this, computational models of PHS revealed that the individual thermal contrast predicted the probability of PHS in healthy

individuals. These results highlight the link between PHS and contrast enhancement in the thermosensory system. We were able to provide a clear, concise explanation that is well-established in other perceptual domains for the paradoxical presence of heat during cooling. Given the distinct response to cold and warm in the periphery, thermal contrast enhancement for PHS is likely driven by broadly tuned neurons within the central nervous system, where inhibition of the surround leads to an amplification of perceived temperature at a thermal boundary, much like what is observed within the visual system[6]. A unifying interpretation of this paradox has powerful implications for how we understand and interpret the experience of pain or heat in response to innocuous stimuli. This is of particular importance to clinical populations, who typically experience incongruent temperature or pain sensations more frequently than healthy populations.

## Data availability

The data are publicly available on GitHub (https://github.com/Body-Pain-Perception-Lab/PHS-TemporalContrast) and OSF[19] (doi: 10.17605/OSF.IO/TP2Q7). The results of the bootstrapped simulations are available through the manuscript's Open Science Framework page (https://osf.io/tp2q7/), as they are too large to be stored on GitHub.

## Code availability

The code, results and figures presented in the manuscript are publicly available on GitHub (https://github.com/Body-Pain-Perception-Lab/PHS-TemporalContrast). Statistical analyses for all data were conducted in R-studio (Version 2023.06.2 + 561). Additional packages used for statistical analyses were *lme4*[20] *lmerTest*[21]*, ROCR*[22]*, boot*[23]*, rsample*[12] *DHARMa*[24], *ggeffects*[25], and *caret*[26].

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

## Acknowledgements

We thank Amalie Holm Lund Sørensen, Anna Villaume Stuckert, Bianka Szöllösi, Camile Maria Costa Correa, Dora Veraszto, Dunja Paunovic, Katarina Vulic, Laura Gaini, Magnus Knudsen, Melina Vejlø, Povilas Tarailis and Sara Viuf for their help with data collection. We also thank Signe Kirk Brødbæk and Thea Rolskov Sloth for data preparation. This publication is based upon work from and supported by the COST Action CA18106 (European Cooperation in Science and Technology). A.G.M., C.S.D., J.F.E. and F.F. are supported by a European Research Council Starting Grant (ERC-2020-StG-948838). M.G.A. is supported by a Lundbeckfonden Fellowship (R272-2017-4345), and M.G.A. and A.S.C. are supported by a European Research Council Starting Grant (ERC-2020-StG-948788). The funders had no role in study design, data collection and analysis, decision to publish or preparation of the paper.

## Author contributions

Conceptualisation: F.F. and K.S. Methodology: F.F. and K.S. Investigation: F.F., A.G.M., M.B., A.S.C., J.F.E., C.S.D., R.A.B. and K.S. Data analysis: A.G.M., M.B., J.F.E. and F.F. Visualisation: A.G.M., F.F. and M.G.A. Writing—original draft: A.G.M., F.F. and M.G.A. Writing—review & editing: A.G.M., F.F., M.G.A., A.C., C.S.D., M.B., J.F.E. and K.S. Supervision: F.F., M.G.A. and K.S.

## Competing interests

The authors declare no competing interests.
