## [Peer Review File · Communications Psychology]

17th Nov 23

Dear Dr Mitchell,

Thank you for your patience during the peer-review process. Your manuscript titled "Temporal Contrast Enhancement in Thermosensation: A Framework for Understanding Paradoxical Heat Sensation" has now been seen by 3 reviewers, and I include their comments at the end of this message. They find your work of interest, but raised some important points. We are interested in the possibility of publishing your study in Communications Psychology, but would like to consider your responses to these concerns and assess a revised manuscript before we make a final decision on publication.

We therefore invite you to revise and resubmit your manuscript, along with a point-by-point response to the reviewers. Please highlight all changes in the manuscript text file.

Editorially, we consider that the following points raised by the reviewers are priority:

1) Reviewers raise concern the modelling of random intercept only in the multilevel analysis. This is a very important aspect that will need to be carefully addressed through a re-analysis. Reviewer #2 has several concrete suggestions on how to implement the GLMER models.

2) Please report full statistics, as per our statistical guidelines (<https://www.nature.com/commspsychol/submit/submission-guidelines#statistical-guidelines>) which are also explained in the formatting checklist and template that are linked below. Reviewer #2 highlights that the DF and their corresponding statistical value (z, t, or F) are missing. Please also verify that GLMER and LMER are correctly applied to binary and continuous outcomes respectively. Reviewer# 2 has several suggestions about this point.

3) Reviewer #2 raise several concerns of the McNemar's test, please verify if this test is the most appropriate for your data, and if this is the case add the relevant information suggested by reviewer #2.

4) Reviewer #1 thinks that the manuscript would benefit from an explanation of the neurophysiological bases of the underlying the Paradoxical Heat Sensation. Please address this aspect in the discussion.

5) Please elaborate the relevance of these results at the application level as suggested by reviewer#3.

6) Please add more details in the methodological section. Reviewer#3 asks for a picture or a detailed description of where the electrodes where the stimulator were placed on the skin, to better specifying the participants' inclusion criteria and explaining how this task differs from those previously used to explore the PHS.

We believe that addressing these points will considerably increase the interpretability of the findings and the impact of this manuscript.

Please note that your revised manuscript must comply with our formatting and reporting requirements, which are summarized on the following checklist: Communications Psychology formatting checklist and also in our style and formatting guide Communications Psychology formatting guide .

Please use the following link to submit your revised manuscript, point-by-point response to the referees' comments (which should be in a separate document to any cover letter) and the completed checklist:

[link redacted]

Please do not hesitate to contact me if you have any questions or would like to discuss these revisions further. We look forward to seeing the revised manuscript and thank you for the opportunity to review your work.

Best regards,

Eva R. Pool

Eva R. Pool, PhD
Editorial Board Member
Communications Psychology
orcid.org/0000-0001-5929-1007

EDITORIAL POLICIES AND FORMATTING

Editorial Policy: Policy requirements (Download the link to your computer as a PDF.)

* **CODE AVAILABILITY:** All Communications Psychology manuscripts must include a section titled "Code Availability" at the end of the methods section. In the event of publication, we require that the custom analysis code supporting your conclusions is made available in a publicly accessible repository; at publication, we ask you to choose a repository that provides a DOI for the code; the link to the repository and the DOI will need to be included in the Code Availability statement. Publication as Supplementary Information will not suffice. We ask you to prepare code at this stage, to avoid delays later on in the process.

* **DATA AVAILABILITY:**

All Communications Psychology manuscripts must include a section titled "Data Availability" at the end of the Methods section or main text (if no Methods). More information on this policy, is available at <http://www.nature.com/authors/policies/data/data-availability-statements-data-citations.pdf>.

At a minimum the Data availability statement must explain how the data can be obtained and whether there are any restrictions on data sharing. Communications Psychology strongly endorses open sharing of data. If you do make your data openly available, please include in the statement:

We recommend submitting the data to discipline-specific, community-recognized repositories, where possible and a list of recommended repositories is provided at <http://www.nature.com/sdata/policies/repositories>.

If a community resource is unavailable, data can be submitted to generalist repositories such as figshare or Dryad Digital Repository. Please provide a unique identifier for the data (for example a DOI or a permanent URL) in the data availability statement, if possible. If the repository does not provide identifiers, we encourage authors to supply the search terms that will return the data. For data that have been obtained from publicly available sources, please provide a URL and the specific data product name in the data availability statement. Data with a DOI should be further cited in the methods reference section.

REVIEWERS' EXPERTISE:

Reviewer #1 Perception of Pain

Reviewer #2 Statistical Analysis

Reviewer #3 Body perception

REVIEWERS' COMMENTS:

Reviewer #1 (Remarks to the Author):

This is an elegant, novel, well-designed and well conducted study on Paradoxical Heat Sensation (PHS). In their manuscript, the authors show that PHS is generated by thermal contrast. This brings a novel way of conceiving thermosensation and PHS. The results are novel, convincing and the study is well powered. I also note the inclusion of male and females in similar proportions, allowing for generalization of the results. The results are of great interest for the fields of sensory systems and neuroscience in general. I also appreciate the clear reporting of analysis methods and statistics, providing the required information to replicate findings. Generally, I greatly appreciated the study and manuscript.

I have only one comment for the authors to consider. I appreciate the conceptual parallel used between the visual and somatosensory systems to explain the role of contrast in PHS. I also appreciate that this is a psychophysical study. However, I think that a point of discussion is missing to strengthen the manuscript on the underlying neurophysiological bases of the PHS. The visual system has been studied in great details to understand contrast mechanisms and visual neurons have been described along the visual pathways to support perceptual finding. For example, the well defined ON center/OFF surrounding receptive fields are very important features of visual neurons. Somatosensory neurons that encode intensity are well described and it would be important to make an attempt at providing a neurophysiological substrate for the observed perceptual results as future directions. I think the authors are missing an opportunity to propose a mechanism here. This should be included in the discussion at least briefly.

Reviewer #2 (Remarks to the Author):

Dear Editor, Dear researchers,

The following is a review of the paper "Temporal Contrast Enhancement in Thermosensation: A Framework for Understanding Paradoxical Heat Sensation". Conceptually the research question and the study design are straightforward and extremely clear. The paper is to-the-point and generally well written. I am not an expert on the domain of heat sensation, however, so have limited knowledge to comment on content. My profession is primarily in statistics, so this is what I chose to focus on in my review, and there are unfortunately many oddities and ambiguities in the current version that require a major revision to be resolved.

DATA ANALYSIS POINTS

1) It is not entirely clear what is meant with "prevalence and rate" (of PHS). Are these two distinct outcomes or interchangeable synonyms for the same outcome? If so, one term only is recommended, or the term should be linked to the analysis. For grouped count data (e.g., in

frequency tables analysed with chi-square tests), it is appropriate to say you analyze rates/frequencies/counts of PHS. For ungrouped count data (e.g., binary outcome in a trial-level GLMER), it is more appropriate to say you analyze the probability/log-odds of a single PHS occurring versus not occurring.

2) None of the reported tests in the manuscript include degrees-of-freedom, which must be reported for chi-square tests and t-tests. Especially for t-tests extracted from a multilevel regression, these DFs can give an indication of effective sample size and model complexity. When using lme4 in conjunction with lmerTest, for example, DFs for F- and t-tests should always have fractional numbers in the numerator (due to the Kenward-Roger or Sattethwaite approximations). For z-tests, there are no DFs (since assuming asymptotic large-sample normality).

3) The section where R software is cited is rather scarce, and probably should be expanded to include other important packages that were used in the analysis. I checked on GitHub and the list is quite a bit longer than reported (but okay for not citing packages that are used only for pre-processing, plotting, etc).

4) It's not clear to me whether McNemar's test was applied correctly here, and Cohen's G is definitely not an appropriate effect size for this test. McNemar's test requires a 2x2 frequency table based on a paired categorical sample (e.g., same categorical response measured twice). Arguably this could be the case e.g., for looking at PHS frequency between the low and medium contrast condition:

MED
LOW PHS-no PHS-yes
PHS-no 45 33
PHS-yes 7 68

Here you could use McNemar's test to analyze whether the marginal PHS distribution shifts between the LOW and MEDIUM contrast condition, and then likewise for MEDIUM to HIGH (or LOW to HIGH). Essentially it is a proportion/binomial test on the table's off-diagonal elements. But is this really what the authors did? What is notably absent from this approach is an omnibus (interaction) test of CONTRAST x TASK on PHS frequency. This could have been accommodated in a multilevel logistic regression of the type reported later in the McNemar paragraph (models 1A and 1B). For trial-level data one could have fitted:

$PHS \sim CONTRAST*TASK + Trial + (1+CONTRAST*TASK+Trial|ID), \text{family=binomial}$

With PHS=0/1 for absence/presence of PHS in a given trial-window. The interaction test (e.g., using car::Anova for the ANOVA breakdown in R) in this model would correspond conceptually to the type of chi-square analysis ordinarily run on a table like e.g.:

TASK
CONTRAST Inn Nox
Low 23 17
Medium 44 39
high 51 43

And follow-up analyses could be run by subsetting data to TASK conditions, or decomposing the model into subtests using R packages like emmeans. The model I suggest above has other additions that the authors should consider, **(a)** a trial number effect to account for adaptation over time, and **(b)** random slopes for all within-subjects effects, to allow individual differences in contrast, task, and adaptation effects. The latter must surely play a role since I expect subjects became either more or less sensitized to heat and pain over the duration of the experiment. In general, separating individual differences from population-effects is always a good idea, and should be done whenever the data and model allow it.

5) Regarding the reported GLMERs, several things confuse me, however. In the McNemar paragraph, the tests for the reported models are t-tests (without DFs), which one normally only obtains from LMERs with a continuous outcome. Where do these t-values come from?

For the analysis on TSL thresholds, (models 2A and 2B) I am not sure why a GLMER was fitted at all, since I assume the threshold is a continuous outcome. Since t-tests are reported, I assume the authors did in fact fit a LMER, although the appendix describes them as mixed logistic regression models. For model 2C, no test statistic is reported (z or t?), only a p-value, while the odds ratios seem to contradict the direction of the written interpretation. For a positive association, $OR > 1$, while for a negative association, $OR < 1$.

For models 3A to 3D, test statistics are again missing when effects are reported.

CONCLUSIONS

In sum, these analyses need to be clarified and possibly re-run based on my suggestions. Some action points to consider for the authors:

- Improve reporting with correct test statistics, degrees of freedom (where applicable), and correct effect sizes. For odds ratios, make sure that the OR follows the direction of the coefficient interpretation.
- Verify if McNemar's test was appropriate for your data, and if that analysis is not better subsumed in a GLMER with a CONTRAST \times TASK design. If you continue to report McNemar's test, the appropriate effect size is the odds of the two off-diagonal 2×2 frequencies.
- Verify for each analysis whether the outcome variable warrants an LMER (continuous outcome) or a GLMER for binomial family (binary outcome). For the former, reporting F- and t-tests is correct, for the latter, reporting Chi-square and z-tests is correct.
- Check if there is evidence for adaptation effects over trials, and for individual differences in the design effects, by including trial number in the model, and random slopes for all within-subjects effects.

Best,
Reviewer

Reviewer #3 (Remarks to the Author):

Main claim:

In this work, Mitchell and colleagues proposed a framework to understand the Paradoxical Heat

Sensation (PHS) phenomenon. To achieve their goal, they applied contrast enhancement principles, known for their instrumental role in understanding visual illusions, to the domain of thermosensation. Their data revealed that in healthy subjects, thermal contrast predicts the occurrence of PHS.

I congratulate the authors because the manuscript is clear, well written and elegant. It is complete with all the information. However, I suggest some implementations in specific points (see below). The work is of interest and worth attention, also to others in the field. Its power resides in the theoretical framework and experimental approach, taking advantage of the paradigm used in the "vision field" and may open new perspectives on the treatment of patients with PHS. The claims are convincing and supported by rigorous methods.

Materials and methods:

- I suggest implementing some information on the experimental paradigm to increase its potential reproducibility (e.g., a photo of the exact position where the stimulator is placed on the skin might help).

- I suggest better specifying the participants' inclusion criteria. For example, I imagine that people with peripheral neuropathy were excluded, and methodological precautions were used before administering the experiment (such as the absence of cosmetic creams in the area of the stimulator application).

- It would be helpful to provide a more explicit justification for the importance of having two innocuous and noxious PHS conditions.

- Furthermore, it would be appropriate to highlight how this task differs from those previously used to explore the PHS.

Statistical analyses:

- I suggest showing information on the statistical assumptions, allowing the use of the approach applied to the study (e.g., distribution of the DVs, etc.).

- Furthermore, in the mixed models, I note that the authors only included the random intercept on the participant ID. Could you please justify your choice?

- As there is no a priori power analysis to establish the sample size tested, it would be appropriate to show the statistical power achieved with 208 participants (e.g. Giner-Sorolla 2019).

- Lastly, for the sake of completeness, I would check in the analysis for the effect of age and gender, which could influence thermal perception.

Moving the knowledge forward:

- The result shown in this paper has relevance on both a theoretical and application level, but I feel that the authors have underdeveloped the latter. For example, how could patients suffering from PHS benefit from this finding?

REVIEWERS' COMMENTS:

Reviewer #1 (Remarks to the Author):

This is an elegant, novel, well-designed and well conducted study on Paradoxical Heat Sensation (PHS). In their manuscript, the authors show that PHS is generated by thermal contrast. This brings a novel way of conceiving thermosensation and PHS. The results are novel, convincing and the study is well powered. I also note the inclusion of male and females in similar proportions, allowing for generalization of the results. The results are of great interest for the fields of sensory systems and neuroscience in general. I also appreciate the clear reporting of analysis methods and statistics, providing the required information to replicate findings. Generally, I greatly appreciated the study and manuscript.

I have only one comment for the authors to consider. I appreciate the conceptual parallel used between the visual and somatosensory systems to explain the role of contrast in PHS. I also appreciate that this is a psychophysical study. However, I think that a point of discussion is missing to strengthen the manuscript on the underlying neurophysiological bases of the PHS. The visual system has been studied in great details to understand contrast mechanisms and visual neurons have been described along the visual pathways to support perceptual finding. For example, the well defined ON center/OFF surrounding receptive fields are very important features of visual neurons. Somatosensory neurons that encode intensity are well described and it would be important to make an attempt at providing a neurophysiological substrate for the observed perceptual results as future directions. I think the authors are missing an opportunity to propose a mechanism here. This should be included in the discussion at least briefly.

Thank you for the complimentary feedback about our manuscript. We are very proud of this work and are happy that you were able to resonate with it. We have added a few lines on page 13, para 1, suggesting a possible central mechanism for this effect.

Reviewer #2 (Remarks to the Author):

Dear Editor, Dear researchers,

The following is a review of the paper "Temporal Contrast Enhancement in Thermosensation: A Framework for Understanding Paradoxical Heat Sensation". Conceptually the research question and the study design are straightforward and extremely clear. The paper is to-the-point and generally well written. I am not an expert on the domain of heat sensation, however, so have limited knowledge to comment on content. My profession is primarily in statistics, so this is what I chose to focus on in my review, and there are unfortunately many oddities and ambiguities in the current version that require a major revision to be resolved.

DATA ANALYSIS POINTS

1) It is not entirely clear what is meant with "prevalence and rate" (of PHS). Are these two distinct outcomes or interchangeable synonyms for the same outcome? If so, one term only is recommended, or the term should be linked to the analysis. For grouped count data (e.g., in frequency tables analysed with chi-square tests), it is appropriate to say you analyze rates/frequencies/counts of PHS. For ungrouped count data (e.g., binary outcome in a trial-level GLMER), it is more appropriate to say you analyze the probability/log-odds of a single PHS occurring versus not occurring.

Thank you for pointing this out. After a review of the manuscript, we realised that the terms rate and prevalence were not clearly used or defined throughout. We have gone through and adjusted this, in accordance with the reviewer's recommendations, throughout the manuscript. The changes are highlighted in yellow throughout.

2) None of the reported tests in the manuscript include degrees-of-freedom, which must be reported for chi-square tests and t-tests. Especially for t-tests extracted from a multilevel regression, these DFs can give an indication of effective sample size and model complexity. When using lme4 in conjunction with lmerTest, for example, DFs for F- and t-tests should always have fractional numbers in the numerator (due to the Kenward-Roger or Sattethwaite approximations). For z-tests, there are no DFs (since assuming asymptotic large-sample normality).

We have now reported degrees of freedom in all reported results, where appropriate.

3) The section where R software is cited is rather scarce, and probably should be expanded to include other important packages that were used in the analysis. I checked on GitHub and the list is quite a bit longer than reported (but okay for not citing packages that are used only for pre-processing, plotting, etc).

We have now updated this section (page 15-16, Statistical analysis section) to include all packages used in our statistical analyses

4) It's not clear to me whether McNemar's test was applied correctly here, and Cohen's G is definitely not an appropriate effect size for this test. McNemar's test requires a 2x2 frequency table based on a paired categorical sample (e.g., same categorical response measured twice). Arguably this could be the case e.g., for looking at PHS frequency between the low and medium contrast condition:

MED

LOW PHS-no PHS-yes

PHS-no 45 33

PHS-yes 7 68

Here you could use McNemar's test to analyze whether the marginal PHS distribution shifts between the LOW and MEDIUM contrast condition, and then likewise for MEDIUM to HIGH (or LOW to HIGH). Essentially it is a proportion/binomial test on the table's off-diagonal elements. But is this really what the authors did? What is notably absent from this approach is an omnibus (interaction) test of CONTRAST \times TASK on PHS frequency. This could have been accommodated in a multilevel logistic regression of the type reported later in the McNemar paragraph (models 1A and 1B). For trial-level data one could have fitted:

$PHS \sim CONTRAST*TASK + Trial + (1+CONTRAST*TASK+Trial | ID), family=binomial$

With PHS=0/1 for absence/presence of PHS in a given trial-window. The interaction test (e.g., using `car::Anova` for the ANOVA breakdown in R) in this model would correspond conceptually to the type of chi-square analysis ordinarily run on a table like e.g.:

TASK

CONTRAST Inn Nox

Low 23 17

Medium 44 39

high 51 43

And follow-up analyses could be run by subsetting data to TASK conditions, or decomposing the model into subtests using R packages like `emmeans`. The model I suggest above has other additions that the authors should consider, **(a)** a trial number effect to account for adaptation over time, and **(b)** random slopes for all within-subjects effects, to allow individual differences in contrast, task, and adaptation effects. The latter must surely play a role since I expect subjects became either more or less sensitized to heat and pain over the duration of the experiment. In general, separating individual differences from population-effects is always a good idea, and should be done whenever the data and model allow it.

Thank you for this comprehensive feedback. We have broken this comment down into key components to make it easier to address:

(1) The use of McNemar's test

Chi Squared tests are a classic way of assessing PHS presence during TSL between and within groups (e.g. Vollert et al., 2022) within the thermosensory literature. The use of McNemar's tests in this manuscript are necessary to be able to draw direct comparisons between our study and others. The reviewer's comment made us realise that the language within the manuscript describing the use of this test was not clear, therefore we have also clarified the language within the manuscript (page 16 para 2).

(2) Effect size for McNemar's

We have converted all reported effect sizes for McNemar's tests from Cohen's g to odds ratios. We would be interested in hearing the reviewers perspective on the use of Cohen's g as an effect size for the McNemar's test. Guides for reporting online state that Cohen's g can be used (e.g. <https://yuzar-blog.netlify.app/posts/2022-02-20-mcnemar/>), and it is a direct output of the `mcnemar's` stats package in R. Therefore, we were under the assumption that this was an appropriate measure of effect size for this test. We have, however, taken the advice of the reviewer onboard and changed all of our effect measures.

(3) Using a mixed regression to model interaction effects

The additional mixed-interaction model that the reviewer suggested does provide an alternative, useful analyses. Therefore we have also included an interaction model (`phs ~ contrast*task + (1|subject)`) within hypothesis 1 and run further pairwise comparisons to compliment McNemar's. These comparisons are reported on page 6, para 2.

(4) The inclusion of trial number

The author suggested including trial number as a fixed effect. First, as there are only three experimental trials, the inclusion of trial as a fixed effect produces model convergence issues. We can achieve model convergence when trial number is z-scored and have included this model (`phs ~ contrast*task + trial_z + (1|subject)`) in the supplementary materials. We did not include it in the main analysis for two reasons. (1) The model including trial is a worse fit than the model without trial and this model was not significantly better than the model without trial number and (2) there was no clear effect of trial on PHS, and therefore we did not feel it was appropriate to include this covariate in the main model.

We did, however, include trial in our linear mixed-effect models (Model 2A and 2B) and found significant effects of trial number on both innocuous and noxious TSL thresholds. We added trial to the model in this case, as it is known that TSL thresholds are affected by the task duration. Both innocuous and noxious thresholds increased (temperature decreased) with trial number, which is in the expected direction and fits with your impression that individuals habituate to the task over time. This outcome is reported in the Supplementary Materials.

(5) The inclusion of all fixed effect as random intercepts

We agree that ideally this is the best approach. However, if we do this we run into model convergence issues. Therefore, we reduced our model complexity until we could obtain a good fit, whilst maintaining a random effect of participant ID. This is the case for the model 1A as well, where if any or the fixed effects are added as random intercepts, the model no longer converges.

5) Regarding the reported GLMERs, several things confuse me, however. In the McNemar paragraph, the tests for the reported models are t-tests (without DFs), which one normally only obtains from LMERS with a continuous outcome. Where do these t-values come from? For the analysis on TSL thresholds, (models 2A and 2B) I am not sure why a GLMER was fitted at all, since I assume the threshold is a continuous outcome. Since t-tests are reported, I assume the authors did in fact fit a LMER, although the appendix describes them as mixed logistic regression models.

This comment has made us realise that we need to further clarify the use of logistic vs. linear mixed effect regression models in our manuscript. We have done so and hope that our reporting is now clearer for the reviewer. The appropriate test statistics have also been added, where missing.

To clarify model use here, the reported tests for hypothesis 1 (Models 1A and 1B, the first results paragraph) were not linear, but logistic mixed-effect models to assess PHS count across conditions. These models have now been replaced with a single interaction model you suggested above (now Model 1A), which is also a GLMER. Models 2A and 2B are lmer models, with linear effects as they are assessing the effect of contrast condition on TSL threshold, which is a continuous variable.

For model 2C, no test statistic is reported (z or t?), only a p-value, while the odds ratios seem to contradict the direction of the written interpretation. For a positive association, $OR > 1$, while for a negative association, $OR < 1$.

For models 3A to 3D, test statistics are again missing when effects are reported.

The missing test statistics have now been reported. We have also reported full results for all models presented in the manuscript in the Supplementary Materials.

Thank you for pointing out the confusion with the direction of interpretation. This is due to the fact that 'higher thresholds' in this context actually means *lower* temperatures. I.e. if someone has a higher cold detection threshold, it means that they require lower cold temperatures to detect cold. It is the same for pain. Therefore, because a higher threshold is actually reflected through a lower TSL temperature (and vice versa) then the statistics are reversed, but the quoted interpretation remains the same. We have clarified this in the text (page 6, final para).

CONCLUSIONS

In sum, these analyses need to be clarified and possibly re-run based on my suggestions. Some action points to consider for the authors:

- Improve reporting with correct test statistics, degrees of freedom (where applicable), and correct effect sizes. For odds ratios, make sure that the OR follows the direction of the coefficient interpretation.
- Verify if McNemar's test was appropriate for your data, and if that analysis is not better subsumed in a GLMER with a CONTRAST \times TASK design. If you continue to report McNemar's test, the appropriate effect size is the odds of the two off-diagonal 2 \times 2 frequencies.
- Verify for each analysis whether the outcome variable warrants an LMER (continuous outcome) or a GLMER for binomial family (binary outcome). For the former, reporting F- and t-tests is correct, for the latter, reporting Chi-square and z-tests is correct.
- Check if there is evidence for adaptation effects over trials, and for individual differences in the design effects, by including trial number in the model, and random slopes for all within-subjects effects.

We believe that these comments have all been addressed both in the response and manuscript, and that your suggestions have improved the clarity and reporting of our results. Thank you again for taking the time to provide feedback.

Best,
Reviewer

Reviewer #3 (Remarks to the Author):

Main claim:

In this work, Mitchell and colleagues proposed a framework to understand the Paradoxical Heat Sensation (PHS) phenomenon. To achieve their goal, they applied contrast enhancement principles, known for their instrumental role in understanding visual illusions, to the domain of thermosensation. Their data revealed that in healthy subjects, thermal contrast predicts the occurrence of PHS.

I congratulate the authors because the manuscript is clear, well written and elegant. It is complete with all the information. However, I suggest some implementations in specific points (see below). The work is of interest and worth attention, also to others in the field. Its power resides in the theoretical framework and experimental approach, taking advantage of the paradigm used in the "vision field" and may open new perspectives on the treatment of patients with PHS. The claims are convincing and supported by rigorous methods.

Thank you for your positive feedback and constructive comments on our manuscript. We have addressed your points below and within the manuscript itself.

Materials and methods:

- I suggest implementing some information on the experimental paradigm to increase its potential reproducibility (e.g., a photo of the exact position where the stimulator is placed on the skin might help).

We have added further details within the manuscript to clarify the experimental design with respect to thermode placement and trial order (page 15, para 2). As the placement of the thermode changed every 6 trials, to reduce carry over effects, a figure would not provide any further clarity than what is already written in the text.

- I suggest better specifying the participants' inclusion criteria. For example, I imagine that people with peripheral neuropathy were excluded, and methodological precautions were used before administering the experiment (such as the absence of cosmetic creams in the area of the stimulator application).

We have stated in the text that participants were healthy, with no underlying pain or neurological conditions, which means individuals with peripheral neuropathy were also excluded. We have also clarified that checks were put in place to ensure that the stimulation site (dorsal forearm) was unaffected prior to the experiment (page 14, para 1).

- It would be helpful to provide a more explicit justification for the importance of having two innocuous and noxious PHS conditions.

Thank you for pointing this out. We have added this clarification to the manuscript on page 5, para 2. We included both these tasks as we wanted to be able to distinguish between PHS elicited from innocuous stimuli and noxious stimuli. This is because there is typically no distinction made between the two categories of PHS within the literature, and we wished to explore the possible differences or similarities between the presence of PHS from both task types. For example, prior to this study it was unclear whether PHS is a consequence of a change to innocuous or noxious TSL thresholds - as this is the first time that the task has been divided in such a way.

- Furthermore, it would be appropriate to highlight how this task differs from those previously used to explore the PHS.

We have now clarified how our task differs from the classic TSL within the manuscript, this can be found on page 14, para 4. To clarify here, our task differs in two ways from the standard TSL. The first, is that we fixed our peak temperatures for each trial to 32, 38 or 44°C. In the standard TSL paradigm these peak temperatures are not fixed, and are instead defined by the temperature at which participant's detected a change in the heating phase, for each trial. Second, for cooling trials we included a tone to indicate when the probe had reached 32°C. This is not in the standard paradigm, and was added to ensure that the

probe was in the cooling phase (below baseline) before participants judged a temperature change.

Statistical analyses:

- I suggest showing information on the statistical assumptions, allowing the use of the approach applied to the study (e.g., distribution of the DVs, etc.).

We have now included distribution plots of TCF contrasts, as well as the log₁₀ of these values, for each condition within the supplementary materials. As the TCF values are a standardised version of the TSL threshold values, the distribution of the TSL thresholds are the same as the TCF, just over a different scale. Therefore, we did not include both TSL thresholds and TCF in this figure. Note the changes in distribution to a more normal scale for the log₁₀ TCF values, which is one of the reasons we chose to use log₁₀ of TCF for models 3B - 3D.

- Furthermore, in the mixed models, I note that the authors only included the random intercept on the participant ID. Could you please justify your choice?

This is a fair comment that was also raised by reviewer two (see comment 4 and our response). We only included the random intercept of participant ID because if we included random intercepts on the full model (with all fixed effects) the model did not converge. The models reliably converged with only subjects included as a random effect. As the highest random variance is explained by participant ID, we chose to keep this variable within the model, and ran the most complex models that produced good fits from our data.

- As there is no a priori power analysis to establish the sample size tested, it would be appropriate to show the statistical power achieved with 208 participants (e.g. Giner-Sorolla 2019).

We have included a post-hoc power calculation on page 14, para 2. We chose to conduct this power calculation based on a logistic regression with a predictor with a lognormal distribution, as this reflects the best fitting model (3D) in our final hypothesis.

- Lastly, for the sake of completeness, I would check in the analysis for the effect of age and gender, which could influence thermal perception.

Models with age and gender as predictors have been added to the supplementary materials. We conducted these analyses on Models 2A, 2B and 3D as we agree with the reviewer and believe it is interesting to look at the effects of age and gender on both TSL temperatures as well as PHS. There was, however, no significant effect of age or gender on

either TSL thresholds or PHS. This is surprising, given previous results have found effects of both age and gender on PHS. The lack of an age effect, however, might be related to the biasing of our sample towards a younger, undergraduate population.

Moving the knowledge forward:

- The result shown in this paper has relevance on both a theoretical and application level, but I feel that the authors have underdeveloped the latter. For example, how could patients suffering from PHS benefit from this finding?

Our work here suggests that healthy individuals are prone to PHS under certain thermosensory conditions, therefore that PHS may be a feature of the thermosensory system and not something that is attached to a specific condition, per se. Understanding the conditions that may produce PHS provides a clearer insight into the mechanisms that drive the presence of PHS in individuals with neuropathic pain, for example. We have clarified this within the manuscript on page 13, para 1, however we do not wish to speculate too much about the mechanisms in patient populations, as this work is conducted solely on healthy individuals. We are currently working to follow up this work in patient and older adult populations, where we will be able to draw more concrete conclusions about how contrast enhancement may elucidate specific PHS mechanisms in patient populations.

25th Jan 24

Dear Dr Mitchell,

Thank you for your patience during the peer-review process. Your manuscript titled "Temporal Contrast Enhancement in Thermosensation: A Framework for Understanding Paradoxical Heat Sensation" has now been seen by 2 reviewers, and I include their comments at the end of this message. They are satisfied with the revised manuscript, but reviewer#2 has still some remaining suggestions. We are interested in the possibility of publishing your study in *Communications Psychology*, but would like to consider your responses to these concerns and assess a revised manuscript before we make a final decision on publication.

We therefore invite you to revise and resubmit your manuscript, along with a point-by-point response to the reviewer. Please highlight all changes in the manuscript text file.

Editorially, we would like to see the notation of the degree of freedom modified as suggested by the reviewer and the concern about the omnibus tests addressed.

I am attaching an Editorial Requests Table that details critical reporting requirements for the revised manuscript. Please attend to each item and ensure your manuscript is fully compliant. We are requesting that your manuscript aligns with these requirements as this facilitates the evaluation of your manuscript, reducing delays in re-review and potential future acceptance. If your revised manuscript is not aligned with these requests on major issues, such as those concerning statistics, it may be returned to you for further revisions without re-review. Additional information can be found in our style and formatting guide Communications Psychology formatting guide.

Please use the following link to submit your

- revised manuscript,
- point-by-point response to the referees' comments,
- cover letter (as a separate document),
- the Editorial Policy Checklist (see below),
- the Reporting Summary (see below), and
- the completed Editorial Request Table (attached):

[link redacted]

We hope to receive your revised paper within 8 weeks; please let us know if you aren't able to submit it within this time so that we can discuss how best to proceed. If we don't hear from you, and the revision process takes significantly longer, we may close your file. In this event, we will still be happy to reconsider your paper at a later date, provided it still presents a significant contribution to

the literature at that stage.

Best regards,

Eva R. Pool

Eva R. Pool, PhD
Editorial Board Member
Communications Psychology
orcid.org/0000-0001-5929-1007

REVIEWER EXPERTISE:

Reviewer #1 Pain perception
Reviewer #2 Statistical analysis

REVIEWER REPORTS:

Reviewer #2 (Remarks to the Author):

Dear authors,

This is a re-review of the paper investigating paradoxical heat sensation. My first round of comments concerned primarily statistical issues and clarifications, and I thank the authors for addressing these thoroughly. It's now much clearer to me which models were run and under which data format and assumptions (e.g., trial-data versus aggregated rates, continuous versus categorical outcomes). I have 4 general comments left for this, of which only the fourth is really critical:

1) COHEN'S G

I realized that I myself had misunderstood the effect size Cohen's G. I had not seen this effect size for a McNemar test before and mistook it for Cohen's d or Hedges' g. So Cohen's G is fine (thanks for the info) and in fact the odds ratios for the McNemar tests can be deleted, since an OR is not really informative for a McNemar test. It would only express the consistency with which participants keep giving the same response (like a repeated measures correlation). For the GLMERs on the other hand, the ORs are appropriate as an effect size.

2) REPORTING DEGREES OF FREEDOM

I thank the authors for reporting all DFs where necessary although an oddity remains regarding the

formatting. For a chi-square test, the convention would be e.g., $\chi^2(1) = x.xx$, $p = x.xxx$. For a t-test in an LMER, the convention would be $t(x.xx) = x.xx$, $p = x.xxx$. So the DF is always put in brackets next to the test statistic letter. For the t-tests, I noticed that the manuscript includes two values. Are these both the corrected and uncorrected DFs? If so, reporting only the corrected DFs suffices.

3) MORE RANDOM EFFECTS

Alright, I take note of the findings regarding additional random effects (e.g. trial number). I do remark that non-convergence is not necessarily problematic for a GLMER/LMER and is very likely to happen for increasingly complex random effects. It does not invalidate the model's estimates, which will often be negligibly close to what it would have been under convergence. The non-convergence warning is different from those citing negative eigenvalues or non-positive definite Hessians, which truly reflect redundancies in the random effects.

4) OMNIBUS TESTS

One important result that still seems to be missing are appropriate omnibus tests. For example, on P6 of the marked-up revised manuscript, it is stated:

"In addition to this, a mixed effect regression model testing the probability of PHS in each trial showed significant interaction effects between task and contrast condition"

But the test result for the interaction is never reported. The authors only report pairwise contrasts and the supplementary material only prints the raw parameter output for each model, not the ANOVA table (which again, only concerns pairwise contrasts). This is not only important for interaction effects, but also for any effect involving "contrastCondition", since it is a 3-level factor, and its overall effect can only be captured with a chi-square/likelihood ratio test (for a GLMER) or an F-test (for an LMER).

Critically, one normally proceeds with pairwise follow-ups only when the omnibus test is significant (to preserve false-positive rate). I believe extracting these results and reporting them should not be too much work. In R it's as straightforward as:

```
library(car)
Anova(model,type=2)
```

Which works for both GLMER and LMER models. Continuing on this point, I would say that printing a model's raw parameter output is not always entirely helpful, in that it will only contain some contrasts but not all of them (e.g., 38-44 contrast). Better probably to print a combination of an ANOVA table (for general effects) with all relevant pairwise contrasts (for specific pairwise comparisons).

CONCLUSION

Other than that, I do not have any further comments on this manuscript, but I do believe the issue of omnibus tests should be addressed. Best of luck with the final revisions!

Kind regards,

Reviewer #2

Reviewer #3 (Remarks to the Author):

I am completely satisfied with the authors' approach to the reviewers' suggestions and the manuscript implementations.

EDITORIAL POLICIES

We ask that you ensure your manuscript complies with our editorial policies and reporting requirements.

To that end, we require revised manuscripts to be accompanied by two completed items: a reporting summary that collects information on study design and procedure, and an editorial policy checklist that verifies compliance with all required editorial policies.

- Nature Research Reporting Summary
- Editorial Policy Checklist

All points on the policy checklist must be addressed. Your revised manuscript can only be sent back to the referees if these checklists are completed and uploaded with the revision.

Notes: If you have submitted a Stage 1 Registered Report, Review, Primer, Comment, or Perspective you do not need to submit these forms. If you have already submitted these forms, you may disregard this request.

* TRANSPARENT PEER REVIEW: Communications Psychology uses a transparent peer review system. This means that we publish the editorial decision letters including Reviewers' comments to the authors and the author rebuttal letters online as a supplementary peer review file. However, on author request, confidential information and data can be removed from the published reviewer reports and rebuttal letters prior to publication. If your manuscript has been previously reviewed at another journal, those Reviewers' comments would not form part of the published peer review file.

Communications Psychology is committed to improving transparency in authorship. As part of our efforts in this direction, we are now requesting that all authors identified as 'corresponding author' create and link their Open Researcher and Contributor Identifier (ORCID) with their account on the Manuscript Tracking System prior to acceptance. ORCID helps the scientific community achieve unambiguous attribution of all scholarly contributions. You can create and link your ORCID from the home page of the Manuscript Tracking System by clicking on 'Modify my Springer Nature account' and following the instructions in the link below. Please also inform all co-authors that they can add their ORCIDs to their accounts and that they must do so prior to acceptance.
<https://www.springernature.com/gp/researchers/orcid/orcid-for-nature-research>

If you experience problems in linking your ORCID, please contact the Platform Support Helpdesk.

Communications Psychology Reviewer response – revision 2

REVIEWER EXPERTISE:

Reviewer #1 Pain perception

Reviewer #2 Statistical analysis

REVIEWER REPORTS:

Reviewer #2 (Remarks to the Author):

Dear authors,

This is a re-review of the paper investigating paradoxical heat sensation. My first round of comments concerned primarily statistical issues and clarifications, and I thank the authors for addressing these thoroughly. It's now much clearer to me which models were run and under which data format and assumptions (e.g., trial-data versus aggregated rates, continuous versus categorical outcomes). I have 4 general comments left for this, of which only the fourth is really critical:

1) COHEN'S G

I realized that I myself had misunderstood the effect size Cohen's G. I had not seen this effect size for a McNemar test before and mistook it for Cohen's d or Hedges' g. So Cohen's G is fine (thanks for the info) and in fact the odds ratios for the McNemar tests can be deleted, since an OR is not really informative for a McNemar test. It would only express the consistency with which participants keep giving the same response (like a repeated measures correlation). For the GLMERs on the other hand, the ORs are appropriate as an effect size.

Thank you for this, we have now replaced all our effect sizes to Cohen's G in the manuscript. These changes are highlighted on page 10.

2) REPORTING DEGREES OF FREEDOM

I thank the authors for reporting all DFs where necessary although an oddity remains regarding the formatting. For a chi-square test, the convention would be e.g., $\chi^2(1) = x.xx$, $p = x.xxx$. For a t-test in an LMER, the convention would be $t(x.xx) = x.xx$, $p = x.xxx$. So the DF is always put in brackets next to the test statistic letter. For the t-tests, I noticed that the manuscript includes two values. Are these both the corrected and uncorrected DFs? If so, reporting only the corrected DFs suffices.

Thank you for highlighting this error. We have now reported the dfs correctly and only included the corrected dfs in the t-tests. These changes are highlighted on page 10.

3) MORE RANDOM EFFECTS

Alright, I take note of the findings regarding additional random effects (e.g. trial number). I do remark that non-convergence is not necessarily problematic for a

GLMER/LMER and is very likely to happen for increasingly complex random effects. It does not invalidate the model's estimates, which will often be negligibly close to what it would have been under convergence. The non-convergence warning is different from those citing negative eigenvalues or non-positive definite Hessians, which truly reflect redundancies in the random effects.

We thank the reviewer for this comment as we were not aware that non-convergence is not a huge issue with GLMER models. Unfortunately, we also do observe warnings that cite negative eigenvalues when we include more random effects within the model, so believe that our model choice remains the most appropriate.

4) OMNIBUS TESTS

One important result that still seems to be missing are appropriate omnibus tests. For example, on P6 of the marked-up revised manuscript, it is stated:

"In addition to this, a mixed effect regression model testing the probability of PHS in each trial showed significant interaction effects between task and contrast condition"

But the test result for the interaction is never reported. The authors only report pairwise contrasts and the supplementary material only prints the raw parameter output for each model, not the ANOVA table (which again, only concerns pairwise contrasts). This is not only important for interaction effects, but also for any effect involving "contrastCondition", since it is a 3-level factor, and its overall effect can only be captured with a chi-square/likelihood ratio test (for a GLMER) or an F-test (for an LMER).

Critically, one normally proceeds with pairwise follow-ups only when the omnibus test is significant (to preserve false-positive rate). I believe extracting these results and reporting them should not be too much work. In R it's as straightforward as:

```
library(car)
Anova(model,type=2)
```

Which works for both GLMER and LMER models. Continuing on this point, I would say that printing a model's raw parameter output is not always entirely helpful, in that it will only contain some contrasts but not all of them (e.g., 38-44 contrast). Better probably to print a combination of an ANOVA table (for general effects) with all relevant pairwise contrasts (for specific pairwise comparisons).

Thank you for taking the time to outline this clearly. The results of the omnibus tests for each model have been added to the supplementary materials and the relevant statistics have been reported for the quoted interaction effect (now on page 10 of the manuscript). The relevant pairwise comparisons are fully reported within the manuscript.

CONCLUSION

Other than that, I do not have any further comments on this manuscript, but I do believe the issue of omnibus tests should be addressed. Best of luck with the final revisions!

Kind regards,
Reviewer #2

Reviewer #3 (Remarks to the Author):

I am completely satisfied with the authors' approach to the reviewers' suggestions and the manuscript implementations.

14th Feb 24

Dear Dr Mitchell,

Your manuscript titled "Temporal Contrast Enhancement in Thermosensation: A Framework for Understanding Paradoxical Heat Sensation" has now been discussed internally and I am delighted to say that we are happy, in principle, to publish a suitably revised version in Communications Psychology under the open access CC BY license (Creative Commons Attribution v4.0 International License).

We therefore invite you to revise your paper one last time to address a list of editorial requests. At the same time we ask that you edit your manuscript to comply with our format requirements and to maximise the accessibility and therefore the impact of your work.

EDITORIAL REQUESTS:

SUBMISSION INFORMATION:

OPEN ACCESS:

Communications Psychology is a fully open access journal. Articles are made freely accessible on publication under a CC BY license (Creative Commons Attribution 4.0 International License). This license allows maximum dissemination and re-use of open access materials and is preferred by many research funding bodies.

For further information about article processing charges, open access funding, and advice and support from Nature Research, please visit <https://www.nature.com/commspsychol/article-processing-charges>

At acceptance, you will be provided with instructions for completing this CC BY license on behalf of all authors. This grants us the necessary permissions to publish your paper. Additionally, you will be asked to declare that all required third party permissions have been obtained, and to provide billing information in order to pay the article-processing charge (APC).

* TRANSPARENT PEER REVIEW: Communications Psychology uses a transparent peer review system. On author request, confidential information and data can be removed from the published reviewer

reports and rebuttal letters prior to publication. If you are concerned about the release of confidential data, please let us know specifically what information you would like to have removed. Please note that we cannot incorporate redactions for any other reasons.

* CODE AVAILABILITY: All Communications Psychology manuscripts must include a section titled "Code Availability" at the end of the methods section. We require that the custom analysis code supporting your conclusions is made available in a publicly accessible repository at this stage; please choose a repository that generates a digital object identifier (DOI) for the code; the link to the repository and the DOI must be included in the Code Availability statement. Publication as Supplementary Information will not suffice.

* DATA AVAILABILITY:

[link redacted]

Best regards,

Antonia Eisenkoeck

Antonia Eisenkoeck
Senior Editor
Communications Psychology

REVIEWERS' COMMENTS: